# Male Infertility and the Risk of Developing Testicular Cancer: A Critical Contemporary Literature Review

**DOI:** 10.3390/medicina59071305

**Published:** 2023-07-14

**Authors:** Giuseppe Maiolino, Esaú Fernández-Pascual, Mario Alberto Ochoa Arvizo, Ranjit Vishwakarma, Juan Ignacio Martínez-Salamanca

**Affiliations:** 1Department of Medicine and Surgery, Urology Clinic, University of Perugia, 06129 Perugia, Italy; giuseppe.maiolino@studenti.unipg.it; 2LYX Institute of Urology, Faculty of Medicine, Universidad Francisco de Vitoria, 28223 Madrid, Spain; mochoa.arvizo@gmail.com (M.A.O.A.); ranjitkarma@gmail.com (R.V.); 3Department of Urology, Hospital Universitario La Paz, 28046 Madrid, Spain; 4Department of Urology, Hospital Universitario Puerta de Hierro-Majadahonda, Universidad Autónoma de Madrid, 28222 Madrid, Spain

**Keywords:** male infertility, testicular cancer, risk factors, germ cell neoplasia in situ, scrotal ultrasound

## Abstract

*Background and Objectives*: The relationship between male infertility (MI) and testicular cancer (TC) is bilateral. On one hand, it is well-established that patients diagnosed with TC have a high risk of pre- and post-treatment infertility. On the other hand, the risk of developing TC in male infertile patients is not clearly defined. The objective of this review is to analyze the histopathological, etiological, and epidemiological associations between MI and the risk of developing testicular cancer. This review aims to provide further insights and offer a guide for assessing the risk factors for TC in infertile men. *Materials and Methods*: A comprehensive literature search was conducted to identify relevant studies discussing the relationship between MI and the risk of developing TC. *Results*: The incidence rates of germ cell neoplasia in situ (GCNIS) appear to be high in infertile men, particularly in those with low sperm counts. Most epidemiological studies have found a statistically significant risk of developing TC among infertile men compared to the general or fertile male populations. The concept of Testicular Dysgenesis Syndrome provides an explanatory model for the common etiology of MI, TC, cryptorchidism, and hypospadias. Clinical findings such as a history of cryptorchidism could increase the risk of developing TC in infertile men. Scrotal ultrasound evaluation for testis lesions and microlithiasis is important in infertile men. Sperm analysis parameters can be useful in assessing the risk of TC among infertile men. In the future, sperm and serum microRNAs (miRNAs) may be utilized for the non-invasive early diagnosis of TC and GCNIS in infertile men. *Conclusions*: MI is indeed a risk factor for developing testicular cancer, as demonstrated by various studies. All infertile men should undergo a risk assessment using clinical examination, ultrasound, and semen parameters to evaluate their risk of TC.

## 1. Introduction

A bidirectional association between testicular cancer (TC) and male infertility (MI) is widely described in the medical literature. This relationship represents a hot topic for urologists, oncologists and all figures involved both in MI care and the management of patients with TC.

Another significant detail is the number of patients involved in this relationship. On the one hand, testicular cancer, with 74,500 new cases worldwide in 2020, ranks as the 20th most common cancer globally and the leading cancer among men aged 15–44 in Europe. While countries in Northern and Western Europe have seen a plateau in incidence rate, there are significant increases in countries with lower incidence rates in Europe [1]. On the other hand, approximately 8–12% of couples are unable to achieve pregnancy within one year, and in around 50% of these cases, a male factor is identified [2]. This number is expected to rise, considering the latest evidence on trends in human sperm quality and production [3].

TC is a well-known risk factor for male reproductive potential due to several reasons. All cancers may impact the reproductive health of men, and multiple factors are likely involved, including pre-existing defects in germ cells, the systemic effects of cancer, and disturbances in endocrine and immunological systems [4]. However, among cancers, testicular cancer shows the highest impact on semen quality. Oligospermia was found in 52% of pre-treatment semen samples of men with testicular cancer compared to 12–30% of men with other cancers [5]. Moreover, seminomas seem to have a more significant effect on sperm production than non-seminomatous tumors [6], and even benign tumors seem to impair pre-treatment semen parameters [7]. Treatment-related factors of TC contribute to potential damage to male reproductive potential. Surgical damage occurs in both orchiectomy [8,9] and testis-sparing surgery, although a significant postoperative decline of sperm parameters is sporadic in the latter case [7]. Chemotherapy and radiotherapy treatments have a well-defined gonadotoxicity [10,11], with radiotherapy having a more negative effect on fertility compared to chemotherapy [12]. Finally, ejaculation disorders induced by retroperitoneal lymph node dissection should also be considered, although primary and post-chemotherapy bilateral nerve-sparing retroperitoneal lymph node dissection has shown high rates of antegrade ejaculation, approximately 95% and 79%, respectively [13]. Testicular cancer treatments have a cumulative deleterious impact, as patients who undergo surgery plus chemotherapy and radiotherapy showed the lowest values of sperm concentration [14]. Nonetheless, the overall success rate of patients trying to conceive after TC treatments, with or without infertility treatments, is approximately 82% [15]. 

Conversely, although MI is a recognized risk factor for developing TC, the evidence on this “side” of the relationship appears to be less well-defined. The primary objective of this narrative review is to assess the evidence reported in the literature from various perspectives, including histopathological, epidemiological, and etiological studies. Additionally, this review aims to present readers with clinical, imaging, and laboratory findings that can help evaluate the risk of TC in the male infertile population. 

## 2. Methods

A narrative review was conducted through an extensive literature search on PubMed between March and April 2023. The following search terms have been used: “infertility’’ OR ‘‘infertile men” AND ‘‘testis cancer” OR “testicular cancer”. We used the following inclusion criteria: human studies with no limitations based on a study design published in the English language between January 1970 and April 2023. Reference lists of included papers were also hand-searched. Data cited from the works of other authors are used in this review only if the original studies are found and revised or, in case it is impossible to find them, the data have been found in the abstract or almost two authors have cited the same data.

## 3. Histopathological Studies: The First Historical Evidence

The first studies that raised suspicion of an association between MI and the risk of developing TC were histopathology studies on the “abnormal morphology of germ cells” found in biopsies from infertile men [16]. These studies primarily focused on the potential neoplastic development of these cells and the definition of “carcinoma in situ of the testis”, now referred to as Germ Cell Neoplasia In Situ (GCNIS). While it is important to note that GCNIS can also be found in fertile men, the male infertile population was the first population in which GCNIS was defined and discovered. Currently, we know that all testicular germ-cell tumors (except yolk-sac tumors, mature teratoma, and spermatocytic seminoma found in relatively older men) are preceded by the presence of GCNIS cells [17]. 

Focusing on male infertile patients (Table 1), several studies from various countries reported the incidence of GCNIS in testicular biopsies ranging from 0 to 3.5% during the 1970s and 1990s. Some authors even recommended testicular biopsy for patients at risk of germ cell carcinoma, including infertile men [18,19,20,21,22,23,24,25,26,27,28]. Moreover, studies reported in the literature after the 1990s have shown the incidence of GCNIS in testicular biopsies from infertile males to be between 0.54% and 6.3% [29,30,31,32,33,34,35]. 

Given the low level of evidence of these studies (the majority of them are retrospective case series with the exception of one prospective study [27] with a Level of Evidence of 4), there is considerable heterogeneity in the reported incidence of GCNIS due to several confounding factors. These factors include differences in the samples of infertile men studied and the relative presence of specific conditions associated with an increased risk of GCNIS, such as men with a history of cryptorchidism or biopsies from undescended testes. Additionally, different sperm alterations that do not pose a potential risk of GCNIS, such as azoospermic men with germ cell aplasia, known as Sertoli-cell-only syndrome, can contribute to the heterogeneity [17,34]. This heterogeneity might suggest that infertility itself may not be the true risk factor, but rather the conditions associated with infertility. However, many authors have reported the presence of GCNIS in infertile men without any recognized cause of infertility [19,21,24,26,30,33,34]. Most likely, the selection of specific features or criteria for infertile men undergoing testicular biopsy could affect the incidence of GCNIS. In more recent studies, testicular biopsies were performed on selected patients with severe sperm count abnormalities, often including cases of azoospermia during TESE procedures, resulting in varying percentages of GCNIS incidence [30,31,33]. 

The accuracy of testicular biopsies in diagnosing GCNIS should also be considered. While the random biopsy of the testis has been reported as an effective tool for diagnosing GCNIS [36], some authors suggested that false-positive cases may occur in the first phase of GCNIS spreading close to the rete testes which are not usually biopsied [24]. This can lead to the development of testicular cancer during the follow-up of testes with negative biopsies for GCNIS. Furthermore, the use of immunohistochemistry (PLAP, OCT3/4, and c-KIT) in more recent series may increase the likelihood of detecting GCNIS [34] compared to series where the histopathological diagnosis was made using HE staining. Other confounding factors to consider are the ages of the infertile male samples (ranging from 20 to 59 years) and the lifetime risk of testicular cancer in different countries. These factors can further contribute to the heterogeneity observed in the reported incidence of GCNIS.

In their extensive review in 2000, Rørth M. et al. [17] estimated that the prevalence of GCNIS in infertile men is likely not higher than 1%, taking into account the biases in older retrospective series. McLachlan et al., in 2007, reported an incidence of 2.4% of GCNIS in bilateral biopsies from infertile men and suggested that the actual prevalence might be even higher [32]. They proposed that a potential GCNIS in infertile men could progress to TC at an earlier age, implying that a patient could be considered an oncological patient before reaching the age at which he might seek medical advice for infertility issues, possibly in his thirties [32]. 

Finally, there is a lack of comparative studies regarding the relative risk of a GCNIS diagnosis in infertile men compared to the general healthy population. However, two studies provide some insight into the prevalence of GCNIS in infertile males versus healthy populations. In a study conducted by Giwercman et al. in Copenhagen, testicular biopsies from 399 healthy males who died suddenly and subsequently underwent autopsy were analyzed. No cases of GCNIS were found in this analysis, but one man had a history of orchiectomy for GCNIS, suggesting a prevalence of GCNIS in the general population of 1 in 399 (0.25%) [37]. The second study, conducted in Germany, involved bilateral testis biopsies performed on 1388 presumably healthy men who died unexpectedly and underwent autopsy. Histopathology analysis was performed using immunohistochemistry with placental alkaline phosphatase staining. In this study, GCNIS was found in six cases (0.43%), a prevalence consistent with the lifetime risk of testicular cancer in Germany [38]. Therefore, considering the prevalence of GCNIS in healthy populations ranging from 0.25% to 0.43%, it appears that infertile men, as observed in recent series with incidences ranging from 0.54% to 6.3%, have a higher incidence of GCNIS diagnosis. 

## 4. Epidemiological Studies: Another Link between Male Infertility and Risk of TC

Numerous observational studies have been published on the risk of TC among men with infertility (Table 2).

Most of the studies reviewed were case-control studies with a low level of evidence (LE: 3b). However, there were six retrospective cohort studies (LE: 2b), most of which were conducted in the USA [47,48,49,51,52] with only one study from Denmark [44]. These studies primarily focused on the risk of testicular cancer among infertile men, except for two studies [51,52] that also analyzed the risk of other cancers.

The majority of studies found a statistically significant risk of testicular cancer among infertile men when compared to a matched sample of men from the general population or fertile men. The assessment of testicular cancer risk varied across studies, using different measures such as relative risk (RR), odds ratio (OR), or standardized incidence ratio (SIR). The definition of “infertile men” also varied, ranging from men with a period of unprotected intercourse without pregnancy for more than 1 year [40] to men with reported problems with low fertility causing difficulty in conceiving [41]. Some studies classified infertile men based on the number of children [42,43,44,45,46,50], while others defined them as “infertile” if they were part of couples evaluated for fertility problems [44,48]. A few studies considered sperm parameters [47,49,52] or the diagnosis and treatment of male infertility [51]. This wide range of definitions complicates the interpretation of the data. Additionally, confounding factors such as cryptorchidism or specific diagnoses in infertile men, socio-economic status, and different baseline risks of testicular cancer in different countries can introduce bias. 

However, the evidence of an increased risk of TC in infertile men is supported by a recent meta-analysis that included four retrospective cohort studies [44,48,51,52]. The meta-analysis reported a 1.9-fold higher risk of testicular cancer in men with male infertility compared to men considered fertile (pooled OR = 1.91, 95% CI: 1.52–2.42) [53]. 

## 5. Etiological Studies: Testicular Dysgenesis Syndrome (TDS), a Common Etiology for MI and TC

A significant body of research consisting of basic, histological, genetic, and experimental studies has been conducted on testis cancer, male infertility, cryptorchidism, hypospadias, and their interconnections. These studies have revealed a complex network of shared mechanisms and associations. However, conducting an extensive review of all these studies is beyond the scope of this particular review. Instead, we will concentrate on the origins, description, and progression of one prominent endeavor that aimed to consolidate this evidence into a unified clinical concept known as Testicular Dysgenesis Syndrome (TDS).

Based on the data gathered from the referenced studies, which indicate various epidemiological trends such as an increased incidence of testicular cancer, declining semen quality, higher utilization of Assisted Reproductive Technology techniques, and rising frequencies of cryptorchidism and hypospadias, Skakkebaek et al. [54] proposed the hypothesis in 2001 that poor semen quality, testicular cancer, cryptorchidism, and hypospadias could be different manifestations of a single underlying syndrome termed Testicular Dysgenesis Syndrome (TDS).

In their initial attempts to elucidate the shared and fundamental pathogenic mechanism of TDS, the authors put forth the concept of “disruption of embryonal programming”, which could be triggered by exposure to estrogenic and anti-androgenic compounds during the in utero or perinatal period, as evidenced by several studies conducted on male animals. This pathogenic mechanism was derived from the “estrogen hypothesis”, first postulated and published by Sharpe and Skakkebaek in 1993 [55] in *The Lancet*. While the authors of the initial TDS paper dismissed the notion of a common genetic cause, they did not rule out the possibility that genetic factors may modulate the effects of environmental factors, either amplifying or mitigating their impact significantly. Considering the cascade of events underlying sex differentiation and subsequent male fetal development, an early disruption could result in multiple defects. If this disruption affects the differentiation of Sertoli and Leydig cells, both spermatogenesis and testosterone production can be impaired. To explain the presence of testicular cancer within the TDS framework, the authors cited the following evidence: the intrauterine alteration in the differentiation of primordial germ cells gives rise to germ cell neoplasia in situ (GCNIS) cells, the heightened risk of testicular cancer associated with rare genetic abnormalities (e.g., 45X/46XY and androgen insensitivity) characterized by cryptorchidism and hypospadias, the epidemiological association between sperm parameter abnormalities and the risk of testicular cancer, the pre-treatment sperm alterations observed in testicular cancer patients, the well-established link between cryptorchidism and testicular cancer, and separate studies demonstrating associations between perinatal factors (such as low birth weight and year of birth) with testicular cancer, hypospadias, and low sperm counts. 

Furthermore, the authors delineated a spectrum of TDS severity, ranging from a mild form characterized solely by impaired spermatogenesis to a complete and more severe form encompassing impaired spermatogenesis, cryptorchidism, and hypospadias, which aligns with rare genetic abnormalities. Within this spectrum, the risk of testicular cancer increases as the severity of TDS advances [54]. 

In 2003, the research group published a study based on the analysis of testicular biopsies from 20 patients with infertility, hypospadias, and undescended testis. They identified histological signs of testicular dysgenesis, including microliths, Sertoli-cell-only tubules, immature seminiferous tubules with undifferentiated Sertoli cells, and tubules containing carcinoma in situ (CIS) cells, which were associated with the clinical features of TDS. Once again, the authors focused on adverse environmental effects rather than specific gene mutations to explain the origin of the syndrome [56]. In the same year, Sharpe published a review discussing the recent insights regarding the estrogen hypothesis [57]. The following year, an editorial authored by Skakkebaek NE and published in the International Journal of Andrology reported new evidence on TDS [58]. This included epidemiological studies on the association between perinatal factors (such as low birth weight, retained placenta, and low parity) and hypospadias, testicular cancer, and cryptorchidism [59], as well as evidence suggesting a common genetic expression between GCNIS cells and embryonic stem cells [60].

Over the subsequent years, the Danish research group published several reviews and studies on TDS [61,62,63,64,65,66,67], which provided further insights into the syndrome. These included the failure or compensation of Leydig cell function, the reduction of anogenital distance, prostate volume, and seminal vesicle volume, as well as potential alterations in “masculine behaviors”. Moreover, the pathogenic mechanism based on exposure to estrogenic and anti-androgenic compounds was reinforced by new experimental studies on the effects of di(n-butyl)phthalate (DBP) [68,69], and the concept of the “masculinization programming window” was discovered [70]. Additionally, new evidence emerged regarding the influence of genetic polymorphisms on susceptibility to environmental factors, although the results were contradictory [71,72]. One of the recent reviews by Skakkebaek NE et al. dedicated a significant section to the influence of epigenetic factors on male reproductive disorders [73].

While TDS remains the most prominent explanatory model for the association between different clinical entities and their specific epidemiological trends, doubts have emerged regarding its validity. The main criticisms of this model are as follows: most evidence of estrogenic/anti-androgenic effects on reproductive abnormalities is primarily based on experimental animal studies, the recent epidemiological trends of the individual entities included in TDS lack consistency (e.g., the increase in testicular cancer incidence does not correspond to a corresponding increase in hypospadias rates during the same period), the potential explanation for decreased sperm count is derived from lifestyle factors acting in adult life (such as obesity and smoking), and the term “syndrome” may not be appropriate for a condition in which a repeated consistent pattern of clinical features cannot be identified [74]. Furthermore, evidence of an increased risk of male reproductive disorders induced by exposure to environmental chemicals, particularly in the case of testicular cancer, is limited [75,76]. 

## 6. How to Evaluate the Risk of TC in Male Infertile Patients?

In the literature, several clinical, imaging, and laboratory findings have been associated with a higher risk of TC in male infertile men. Familiarity with these features is crucial for evaluating which male infertile patients may benefit from follow-up, biopsy, or specific screening modalities. 

### 6.1. Clinical Findings

Although there are well-defined risk factors associated with TC in the general population [77], certain risk factors can potentially increase the risk of TC in male infertile patients.

Age is one of the principal risk factors for TC, with peak incidence reported in the third and fourth decades of life for different types of TC [78]. It is worth noting that infertility visits also peak in the third decade of life for both sexes [79]. Some authors have suggested earlier screening for male infertility in young adult men to better manage potential causes of male infertility and to encourage them to make lifestyle changes prior to attempting conception [79]. This highlights the importance of considering the risk of TC in male infertile patients based on their age. Additionally, a consultation for infertility in young adult men can provide an opportunity to evaluate not only the cause of infertility but also the risk factors for TC and improve early diagnosis in this high-risk population.

Cryptorchidism, or undescended testes, is the most common congenital genitourinary abnormality [80,81] and is associated with an increased risk of both MI [82] and TC. Both the history (and subsequent treatment) and the presence (untreated) of undescended testes have a higher risk of TC. Two meta-analyses performed in 2004 and 2010 on cohort and case-control studies found similar risks of TC with an overall relative risk of 4.8 (95% CI: 4.0–5.7) and an odds ratio of 4.30 (95% CI: 3.62–5.11), respectively [83,84]. Both meta-analyses underlined the presence of possible biases derived from methodological problems of assessing the history of cryptorchidism and the difficulty of differentiating between true cryptorchidism and other maldescent testes (such as gliding and retractile testes). A subsequent two meta-analyses, both published in 2013, found a pooled relative risk for TC of 2.90 (95% CI: 2.21–3.82) [85] and of 4.1 (95% CI: 3.6–4.7) [86]. Banks K et al., performing a subgroup analysis, found the following as determinants of a stronger association: bilateral cryptorchidism, unilateral cryptorchidism, and ipsilateral TC, delayed treatment, TC diagnosed before 1970, and seminoma histology [86]. Finally, a more strict and recent meta-analysis on congenital cryptorchidism cases that underwent surgery before adulthood (until the age of 20 years) reported a pooled odds ratio for TC of 3.99 (95% CI: 2.80–5.71) [87]. Regarding the modification of the risk when surgical correction of cryptorchidism is performed, the first interesting meta-analysis by Walsh et al. published in 2007 showed an odds ratio of 5.8 (95% CI: 1.8–19.3) for TC in cryptorchidism cases that underwent delayed or unperformed surgical correction (orchiopexy), compared to those in whom it was performed early [88]. In the same year, a wide retrospective cohort study reported that the relative risk of TC among patients who underwent orchiopexy before 13 years of age halved compared to those treated at 13 years of age or older [89]. Controversial data exist about the risk of TC of the descended testes in cases of unilateral cryptorchidism: while a meta-analysis reported in 2009 a pooled relative risk of 1.74 (95% CI: 1.01–2.98) in the contralateral testes among men with a unilateral undescended testis [90], in the same year, a review on assumptions about cryptorchidism rejected this higher risk [91]. No data compared fertile and infertile patients with a history of cryptorchidism to explore a potential difference in their risk of TC, however, all infertile men with a history of cryptorchidism have a very high risk of developing TC.

Another major risk factor for TC is a family history of TC [77]. Del Risco Kollerud R et al. in 2019 found a very high relative risk (6.3–4.4 fold) for brothers, sons, and fathers of patients affected by TC and a relatively high risk for paternal and maternal uncles (2.0–4.4 fold) [92]. Therefore, a familial history of TC in male infertile patients is an important variable to evaluate. Conversely, some data suggest that there is a familial risk of TC in men with poor semen quality. Anderson et al. reported in a retrospective cohort study a 52% increased risk of testicular cancer in first-degree relatives of men with poor semen quality compared with the first-degree relatives of the control fertile population [93]. Despite the few data and low level of evidence, infertile men should be aware of this risk.

Even if few data are found in the literature, individuals born with low birth weight seem to have an increased risk of TC [84] and, at the same time, an increased risk of fertility issues [94,95], although there are no studies on the potentially additional risk of TC in a patient with male infertility and low birth weight.

Among the potential risk factors for TC, as evaluated by Yazici et al. [77] in a recent review, smoking represents an interesting common risk factor for TC and MI. For the risk of MI, smoking represents one of the most well-defined risk factors, with numerous meta-analyses that confirm this association [96,97]. Indeed, a recent meta-analysis on twelve epidemiological studies found a significant summary odds ratio (sOR) of 1.18 (95% CI: 1.05–1.33) between tobacco smoking and TC [98].

Finally, special consideration is necessary for male infertile patients with a history of unilateral TC. As cited above, infertility can result from disease and treatment-related factors and for these reasons, it is not uncommon to perform a consult for infertility in this population of patients. After a unilateral orchiectomy or testis-sparing surgery, the remaining testicular tissue is to be considered at risk of a metachronous contralateral (in the case of unilateral orchiectomy) and/or ipsilateral (in the case of testis-sparing surgery) testicular cancer. Few data are reported about bilateral TC cases that seem to account for 1–5% of all TC [99], and metachronous tumors represent 65% of these cases [100]. In a large population-based cohort of men diagnosed with testicular cancer before the age of 55 (27,870 men), authors found a 15-year cumulative risk of 1.9% of developing a contralateral metachronous TC, with a lower risk if the first histological diagnosis was non-seminomatous [101]. Even less data are reported regarding the testis-sparing surgery, in which the difference between a local recurrence and a new metachronous tumor is difficult to evaluate. Another interesting data to consider are the utilization during the treatment of TC of platinum-based chemotherapeutic agents that, as reported by some authors [101,102], could have a protective effect on the risk of metachronous tumors.

### 6.2. Ultrasound Findings

Although the strength of recommendation is weak, the latest EAU guideline on Male Infertility recommends performing a scrotal ultrasound in patients with infertility due to a higher risk of testicular cancer [78]. Scrotal ultrasound can provide valuable information in infertile men for various reasons, such as measuring testicular volume and detecting patterns, texture, and findings associated with impaired spermatogenesis or indications of obstruction. However, two specific findings are considered essential for assessing the risk of testicular cancer: nodules/small testicular masses and microlithiasis.

#### 6.2.1. Nodules/Small Testicular Masses

Incidentally, palpable nodules and non-palpable small testicular masses detected during scrotal ultrasound (US) in male infertile patients are not uncommon and can arise from various types of lesions. While palpable lesions in fertile men often warrant orchiectomy due to a high likelihood of malignancy [103,104], the management of non-palpable testicular lesions involves different treatment options. These options may include ultrasound follow-up, radical orchiectomy, or testicular exploration, with the final decision based on frozen section analysis results [105,106,107].

Although ultrasound has a high sensitivity (96.6%) for diagnosing small intratesticular masses, its specificity is low (44%), making it difficult to differentiate between benign and malignant lesions [108]. However, the use of color Doppler [109] and multiparametric US [110] can improve specificity.

Managing these masses in infertile men can be challenging as preservation of the potential pool of spermatozoa is desired, avoiding orchiectomy [111]. Evidence regarding these lesions in the literature is highly heterogeneous, with variations in sampled populations, lesion classifications, and management approaches (follow-up or interventions), often relying on retrospective designs. A recent meta-analysis noted a higher incidence of small testicular masses among men seeking consultation for infertility (2.86%) compared to those undergoing US for other indications (1.41%) [112]. Another systematic review reported on 1348 lesions in adult asymptomatic men, both fertile and infertile, with single, incidentally identified small testicular lesions (<2 cm), negative tumor markers, and no specific risk factors for malignancy. The pooled data from 108 infertile cases (77.7%) and 31 fertile cases (22.3%) did not reveal a statistical difference in the frequency of benign and malignant lesions [113].

There are several limitations to these two meta-analyses. First, they include studies with low-evidence quality, such as case reports and congress abstracts [113]. Second, the criteria for defining small lesions differ between the two meta-analyses. In the first meta-analysis, small lesions were defined as “only on US evaluation with less than 1 cm”, while in the second meta-analysis, they were defined as “incidentally identified small testicular lesions < 2 cm […] with negative tumor markers and without specific risk factors for malignancy”.

Furthermore, the analysis does not provide information about the specific population samples in which ultrasound investigations were performed. The review includes studies without a defined population sample, and many studies define the population sampled as having “various indications for US”. These limitations make it difficult to evaluate the incidence of small lesions in a population of men with infertility issues who undergo ultrasound examinations. Additionally, it is challenging to determine the relative incidence of palpable and nonpalpable lesions and the percentage of these lesions that will ultimately result in malignancy, benign tumors, benign lesions, or receive no histological diagnosis due to reasons such as patient refusal or surveillance protocols.

To address these limitations, Table 3 only includes studies from the literature that involve a population of patients with infertility issues. This includes men with a specific diagnosis of infertility, men with abnormal semen parameters, couples diagnosed with infertility, and men attending infertility clinics. These studies report on clinical examinations and/or ultrasound examinations that identified suspected testicular lesions. The table provides information on the percentage of malignancy, benign tumors, benign lesions, and both palpable and non-palpable lesions in patients who were not operated on, considering the presence or absence of other risk factors for testicular cancer.

Most of the studies included in the table are retrospective studies involving a consecutive or non-consecutive series of ultrasound examinations.

Based on the data presented in Table 3, the following findings can be summarized: the incidence of testes lesions in patients with infertility who underwent clinical and scrotal ultrasound exams varied from 0% to 37.7%. Among all the groups, azoospermic men showed the highest incidence of lesions, ranging from 1.42% to 37.7%. It is important to note that the high incidence of 37.7% was reported in azoospermic patients who were examined before undergoing diagnostic or therapeutic testicular biopsy. However, this value may be influenced by selection bias. Another important observation is that non-palpable testicular lesions accounted for 42.9% to 100% of the total lesions found in patients with infertility issues. Among these non-palpable lesions, 66.6% to 100% were incidental findings in patients without any risk factors other than male infertility. Regarding the nature of the lesions, the percentage of malignant lesions was reported to be 50% to 100% in palpable testes lesions and 5% to 55% in non-palpable testes lesions. The only prospective controlled study found in the literature [125], although it had a small sample of ultrasound examinations, reported no cases of TC in either the infertile or control groups.

#### 6.2.2. Microlithiasis

Microlithiasis is recognized as a general risk factor for germ cell neoplasia in situ (GCNIS) and TC. Three meta-analyses have investigated the risk of TC in patients with testicular microlithiasis [127,128,129]. The first meta-analysis concluded that, in the presence of risk factors, testicular microlithiasis in adult patients was associated with a significantly increased risk of concurrent TC or GCNIS [127]. The second meta-analysis found a significant association between testicular microlithiasis and the risk of testicular cancer [128]. In 2015, a subcommittee of ESUR (the European Society of Urogenital Radiology) published a guideline on the management of microlithiasis using ultrasound, stating that annual imaging follow-up was recommended only in the presence of additional risk factors until the age of 55 [130]. The guideline considered risk factors such as personal/family history of TC, genetic disease, maldescent testes, history of orchidopexy, and testicular atrophy, but infertility was not specifically included. The subcommittee mentioned that the additional risk of infertility alone was difficult to determine and did not list it among the risk factors. The third meta-analysis, conducted by Aoun F et al. after these guidelines, focused on microlithiasis with or without other risk factors for TC and concluded that microlithiasis alone, without other risk factors, had a similar risk as the general population. However, male infertility was recognized as a risk factor for TC in conjunction with microlithiasis based on fourteen studies included in their meta-analysis, along with a familial or personal history of testicular cancer, testicular atrophy, hypospadias, and cryptorchidism [129]. Lastly, a recent fourth meta-analysis of case-control studies addressed the risk of TC in infertile men with or without testicular microlithiasis, reporting an 18-fold higher prevalence of testicular cancer in infertile men with microlithiasis [131]. Considering the emerging evidence, it is likely that future guidelines will also recognize male infertility as an additional clear risk factor for microlithiasis detected by ultrasound, requiring close follow-up.

### 6.3. Multiparametric Magnetic Resonance Imaging (mpMRI)

The first use of magnetic resonance imaging (MRI) for pathologies of the scrotum was first reported in the 1990s [132]. In the subsequent years, several reports investigated the use of mpMRI and in 2018, the Scrotal and Penile Imaging Working Group (SPI-WG) of the European Society of Urogenital Radiology (ESUR) published guidelines about the use of MRI in the scrotum imaging workup. The consensus paper does not mention the word “infertility”, although many recommendations could be useful for infertile men: differentiating intratesticular from paratesticular masses and differentiating benign from malignant testis masses (as second-line after an ultrasound exam) and paratesticular masses (in this latter case, probably as first-line since the ultrasound does not always allow for confident characterization) [133]. 

Recently, the use of mpMRI has been found useful for evaluating the spermatogenesis function. A recent interesting meta-analysis by Tsili AC et al. reported that, although the data are still preliminary and heterogeneous, in the future mpMRI could be useful to estimate the morphofunctional state of spermatogenesis in male infertile patients, especially with the ADC sequence and magnetization transfer ratio (MTR) [134,135]. The analysis of metabolites by Magnetic Resonance Spectroscopy appears promising [136,137].

Currently, there are no studies that correlate mpMRI findings with the risk of developing TC in infertile males as, for example, the microlithiasis in the ultrasound exam. The potential use of mpMRI to investigate the severity of impaired spermatogenesis could be used to correlate damage and the risk of developing TC since more severe semen analysis parameters already correlate with a higher risk of developing TC (see the next paragraph). 

## 7. Laboratory Exam: Semen Analysis

Most of the data between semen analysis parameters and the risk of TC in infertile men are derived from epidemiological case-control and retrospective cohort studies (see Table 2). Although there is a high risk of bias, there are many interesting findings in these studies. 

Overall, men are considered infertile in couples evaluated for infertility, although normospermic men have a higher risk of TC if compared with the fertile or general population, and patients with oligo- and azoospermia have an even higher risk [44,47,48,49,52]. 

The risk of TC in oligospermic men ranges from 1.3-fold from studies in Denmark [44] to 12–22 fold [47,52] from studies in the USA [47,52], when compared to controls. The wide difference between these studies can be related to the baseline general population risk (the highest testicular cancer rate is observed in Nordic countries, including Denmark and Norway [138]) and selection bias in the cut-off values used to define oligospermia. Moreover, there appears to be a trend between the severity of oligospermia and the increased risk of TC [52]. Among infertile patients, azoospermic patients showed a 2.2-fold higher TC risk compared to non-azoospermic men [49]. Additionally, azoospermic patients who had fathered children before the diagnostic semen analysis of azoospermia showed a lower risk than azoospermic men without children [44]. Indeed, hyperzoospermic men have a TC risk equivalent to that of the general population [52]. Another interesting finding is that men undergoing vasectomy do not have a significantly higher TC risk when compared with controls [51]. 

Although there are less data about motility and morphology, the decline of these parameters is also reported to be associated with a higher risk of TC. Hanson HA et al. found an increased risk of TC when comparing patients with a semen analysis with fertile control subjects matched by age and birth year, as motility, viability, and TMC declined [52]. 

Finally, there is a trend in which the risk of TC increases when multiple numbers of semen alterations are combined [44].

## 8. Future Directions: Non-Invasive Diagnosis of Preinvasive GCNIS and TC

For patients at high risk of developing testicular cancer (TC), the use of noninvasive semen analysis for the diagnosis of germ cell neoplasia in situ (GCNIS) could be a valuable method, including for infertile men. Various attempts have been made to detect CIS cells in semen, as these cells are shed from the seminiferous tubules into the seminal fluid. Diagnostic techniques such as cytological examination and immunohistochemistry with cytoplasmic and nuclear markers like OCT 3/4, AP-2C, and NANOG have been employed, but they have yielded unfavorable results so far. Although these techniques exhibit a high level of specificity, the relatively high number of false positive results and the labor-intensive procedures hinder their reproducibility in standard clinical practice. For a comprehensive review of these endeavors, we recommend the study by Elzinga-Tinke JE et al., 2015 [139].

Recently, microRNA (miRNA) clusters, specifically miR-371-3 and miR302/367, have emerged as novel serum biomarkers for testicular cancer. miRNAs play a role in regulating mRNA translation through epigenetic processes, influencing cellular differentiation and carcinogenesis. As such, miRNAs can function as oncogenes or tumor suppressors [140]. Several studies have demonstrated that miR-371a-3p has a high sensitivity (86%) and specificity (92%) in detecting testicular germ cell tumors. However, there have been conflicting results regarding the use of miR-371a-3p and related miRNAs in diagnosing GCNIS through serum testing, with expression levels ranging from 0% to 51.9%. This discrepancy may be due to the low expression of these markers in GCNIS, as a positive correlation between the diameter of the primary lesion and miR-371a-3p levels has been observed [141]. Consequently, despite some progress in this field [142], further efforts are needed to develop a noninvasive, highly sensitive, and reliable method for GCNIS detection that can be utilized for routine screening of individuals at risk for TC.

## 9. Conclusions

Male infertility has been established as a well-defined risk factor for testicular cancer. While specific conditions leading to male infertility can either decrease or increase this risk, several studies have demonstrated a general association between male infertility and testicular cancer. This association is supported by common etiopathology findings, histological precancerous associations, and population-based evidence. Therefore, a comprehensive risk assessment is necessary for all patients with male infertility. This assessment can be conducted based on clinical history, ultrasound examination, use of mpRMI, and semen analysis parameters. Implementing a structured follow-up schedule and continuing the development of new tools for the non-invasive diagnosis of testicular cancer and preinvasive lesions would greatly benefit male infertile patients. These advancements would enable early detection and intervention, leading to improved outcomes and management of testicular cancer in this population.

## Figures and Tables

**Table 1 medicina-59-01305-t001:** Studies on incidence of GCNIS in biopsies of infertile men.

Reference	Country (Year)	Sample	Mean (Range) Age	Pre-Biopsy Sperm Alterations	Incidence of GCNIS, % (n/Patients)	Bilateral GCNIS, n/Total GCNIS	Idiopathic Infertile Men, n/Total GCNIS	Known Causes of Infertility, n/Total GCNIS	Method(s) of Diagnosis
From 1970s to 1990s									
Nüesch-Bachmann, I.H et al. [18]	Switzerland (1977)	Testicular biopsy from 1635 infertile men	NR	NR	0.55% (9/1635)	NR	NR	NR	NR
Skakkebaek, N.E et al. [19]	Denmark (1978)	812 consecutive testicular biopsies (unilateral and bilateral) from 555 infertile men	31 (22–50)	Most of the patients had azoospermia or oligospermia	1.1% (6/555)	2 (33.3%)	4 (66.6%)	2 (33.3%)	HE stain
Sigg, C. et al. [20]	Switzerland (1981)	2178 testicular biopsies from infertile men	29.4	NR	0.46% (10/2178)	1	NR	NR	HE, Van Gieson’s stain, and PAS
Pryor, J.P. et al. [21]	London (1983)	Histological material from 2043 male partners of infertile marriages	NR	Azoospermia or severe oligozoospermia, some during varicocele ligation	0.39% (8/2043)	NR	4 (50%)	4 (50%)	HE
Burke, A.P. [22] data reported by [17]	USA (1988)	Testicular biopsies from 381 patients	NF	NF	1.8% (7/381)	NF	NF	NF	NF
Schütte, B [23] data reported by [26]	Germany (1988)	Testicular biopsies from 2047 infertile men	NF	Oligospermia	0.73% (15/2047)	NF	NF	NF	NF
Nistal, M et al. [24]	Spain (1989)	Bilateral testicular biopsies were performed in 723 men consulting for infertility	NR	NR	0.69% (5/723)	0	3 (60%)	2 (40%)	NR
Mougharbel, S. et al. [25] data reported by [17]	France (1990)	Testicular biopsies from 150 infertile men	NF	NF	3.3% (5/150)	NF	NF	NF	NF
Bettocchi, C. et al. [26]	London (1994)	Uni- or bilateral testicular biopsies performed on 2739 infertile men	NR	Azoospermia (with low FSH), persistent severe oligozoospermia, or at the time of varicocele ligation or vasectomy reversal.	0.6% (16/2739)	0	10 (62.5%)	6 (37.5%)	NR
Giwercman, A. et al. [27]	Denmark (1997)	Uni- or bilateral testicular biopsy from 207 consecutive oligozoospermic men from infertile couples	31 (20–45)	Sperm density <10 million/mL or sperm density <20 million/mL, history of cryptorchidism or one or two atrophic testicles, or both. Excluded if: both testes <4 cm^3^ (high probability of Klinefelter’s syndrome); or history of testicular cancer or treated with cytotoxic drugs	0% (0/207)	-	-	-	NR
Zorn, B. et al. [28]	France (1998)	Testicular biopsies from 85 infertile men	NR	Most with diagnosed azoospermia	3.5% (3/85)	NR	NR	NR	NR
After 1990s									
von Eckardstein, S. et al. [29]	Germany (2001)	Testicular biopsies in 76 men with infertility	33,4	NR	2.6% (2/76)	NR	1 (50%)	1 (50%)	NR
Mancini, M. et al. [30]	Italy (2007)	Testicular biopsy in 97 infertile men	34.9	Azoospermic infertile men	2.1% (2/97)	NR	1 (50%)	1 (50%)	NR
Olesen, I.A et al. [31]	Denmark (2007)	Testicular biopsy in 453 infertile men	33.1 (20.7–49.1)	Severity of the impairment of semen quality was considered in indication for biopsy	2.2% (10/453)	3	NR	NR	HE and PLAP
McLachlan, R.I. et al. [32]	Australia (2007)	Bilateral testicular biopsy in 534 infertile men	NR	NR	2.4% (13/534)	0	NR	NR	NR
Negri, L. et al. [33]	Italy (2008)	Testicular biopsies from 370 infertile men	NR	Suspected testicular dysgenesis at ultrasound	0.54% (2/370)	0	0	2 (100%)	HE, in selected cases PLAP and c-Kit
van Casteren, N.J. et al. [34]	Netherlands (2009)	158 testicular biopsies in the context of infertility	NR	NR	6.3% (10/158)	NR	NR	NR	Immunohistochemistry for OCT3/4
Soltanghoraee, H. et al. [35]	Iran (2014)	Uni- and bilateral testicular biopsies of 1153 infertile men	35.3 (21–59)	Azoospermic	0.6% (7/1153)	0	1 (14.3%)	6 (85.7%)	HE and PLAP

GCNIS: Germ Cell Neoplasia in situ; NR: Not Reported; NF: Not Found; HE: Hematoxylin and Eosin; PAS: Periodic Acid–Schiff; PLAP: Placental Alkaline Phosphatase.

**Table 2 medicina-59-01305-t002:** Epidemiological studies on risk of TC in infertile men.

Reference	Type of Study—LE	Sample	Principal Infertility Definition	Principal Result	Comments and Other Results Based on Different Infertility Definitions
Focused on risk of TC					
Henderson, B.E. et al. [39]	Case-control study (USA)—3b	131 cases of TC vs. 131 controls	Married men with no Children	OR = 1.38 (*p* < 0.65)	Authors did not find a significant difference in the risk of TC between married men with children and married men without children
Swerdlow, A.J. et al. [40]	Case-control study (UK)—3b	178 cases of TC vs. 315 controls	Period of unprotected intercourse without achieving pregnancy for more than 1 year	**RR = 1.76** **(95% CI: 1.08–2.86)**	Increased risk for patients with seminoma, but not statistically significant when cryptorchid patients were excluded. Considering only married men, a similar proportion of cases (21%) and control (20%) had not fathered any children or had any other pregnancies Duration from marriage to first delivery was slightly longer for cases compared to controls, but this difference was not statistically significant
UK Testicular Cancer Study Group [41]	Case-control study (UK)—3b	794 cases of TC vs. 609 controls	Reported problems with low fertility causing difficulty in conceiving	OR = 2.66 (95% CI: 0.94–7.54)	Small proportion of cases with reported “low fertility (1.7%)”
Møller, H. et al. [42]	Case-control study (Denmark)—3b	514 cases of TC vs. 720 controls	Men were classified into low, normal, or high fertility, based on the number of expected children compared to controls	**Low fertility—OR = 1.98** **(95% CI: 1.43–2.75)**	After adjusting for confounding factors (cryptorchidism, testicular atrophy, mumps orchitis, age at first intercourse, sexually transmitted diseases, etc.) OR remained statistically significant with an OR of 2.13 (95% CI: 1.51–3.00) There was no protective effect observed with a higher number of children than expected The associations between MI and TC were similar for both seminoma and non-seminoma cases.
Jacobsen, R et al. [43]	Case-control study (Denmark)—3b	3530 cases of TC vs. 1,488,957 controls	Paternal fertility rate prior to the cancer diagnosis	**Standardized fertility rate ratio (in period up to 2 years before cancer diagnosis) = 0.93 (95% CI: 0.89–0.97)**	The paternal fertility ratio was strongly reduced in cases diagnosed with non-seminoma. There was a significantly lower proportion of male offspring (offspring sex ratio) born to the patients compared to the expected value. This observation was consistent for both seminoma and non-seminoma cases.
Jacobsen, R et al. [44]	Retrospective Cohort study (Denmark)—2b	32,442 men (compared with total population of Danish men)	Men in couples with fertility problems	**SIR = 1.6** **(95% CI: 1.3–1.9)**	Oligospermic men were defined as <20 × 10^6^/mL **[SIR = 2.3 (95% CI: 1.6–3.2)]**, poor motility of the spermatozoa [SIR = 2.5 (95% CI: 1.0–5.2)], and high proportion of morphologically abnormal spermatozoa [SIR = 3.0 (95% CI: 0.8–7.6)] were all associated with an increased risk of TC Risk increased with increasing number of subfertility measures [for all three subfertility measures SIR = 9.3 (95% CI: 1.0–33.4)] Azoospermic men who had fathered children before semen analysis [SIR = 2.0 (95% CI: 0.7–4.3)] showed lower risk of TC than azoospermic men without children **[SIR = 3.5 (95% CI: 1.4–7.2)]** when compared with controls
Richiardi, L. et al. [45]	Case control study (Sweden)—3b	4592 cases of TC vs. 12,254 controls	Number of children Unlike-sex twins rate	Number of children (≥3 vs. 0) for TC diagnosis—**OR = 0.71, (95% CI: 0.62–0.81)** Case patients were less likely to be unlike-sex twins than control subjects—OR = 0.49 (95% CI: 0.22–1.08)	It is assumed that male subfertility affects the rate of dizygotic twinning, but not monozygotic twins. To estimate the dizygotic twinning rates, the authors used the unlike-sex twins rate, which is not influenced by decisions about family size. This unlike-sex twins rate was reported as a measure of infertility, comparing the rate between case patients and controls. The authors also found an increased twinning rate after the diagnosis in case patients, likely attributed to treatment for iatrogenic subfertility.
Doria-Rose, V.P. et al. [46]	Case-control study (USA)—3b	329 cases of TC vs. 672 controls	Men with no children	Age-adjusted OR = 0.76 (95% CI: 0.54–1.06)	Inverse associations were seen for seminomas and nonseminomatous TC. Adjusted OR for cryptorchidism (OR = 0.82, 95% CI: 0.58–1.15) suggesting a potential confounding factor There was no evidence indicating that an increasing number of children leads to a further decrease in the risk of TC
Raman, J.D. et al. [47]	Retrospective Cohort study (USA)—2b	3800 infertile men (vs. age matched controls from the general population)	Men evaluated for a male factor infertility with sperm concentration < 20 × 10^6^/mL and motility < 50% or morphology < 30%	**SIR = 22.9** **(95% CI: 22.4–23.5)**	Among infertile men over a 10-year period, only 10 (0.3%) were diagnosed with a testicular cancer, all of which were seminomatous. Excluding the 2 patients with a history of cryptorchidism, they found a decrease of **SIR to 18.3 (95% CI 18.0–18.8).**
Walsh, T.J. et al. [48]	Retrospective Cohort study (USA)—2b	22,562 males (vs. age-matched sample of men from the general population)	Male partners in couples evaluated for infertility	SIR = 1.3 (95% CI: 0.9–1.9)	When compared to the general population, men with “male factor infertility” and abnormal semen analysis variables had an **SIR = 2.8 (95% CI: 1.5–4.8)** Among men without male factor infertility, there was no significant evidence of an increased testicular cancer risk (SIR= 1.1; 95% CI: 0.6–1.7). In couples evaluated for infertility, both with and without a “male factor”, a Cox proportional hazards regression model (controlling for age, duration of infertility treatment, and infertility treatment center) found a **HR = 2.8 (95% CI: 1.3–6.0)**
Eisenberg, M.L. et al. [49]	Retrospective Cohort study (USA)—2b	2238 men (vs. expected number of cases in general population)	Men with complete records evaluated for infertility at a single andrology clinic	**SIR = 1.7** **(95% CI: 1.2–2.5)**	Stratified by azoospermia status, only azoospermic men had a statically significant elevated risk of testicular cancer **(SIR = 2.9, 95% CI: 1.4–5.4)** When considering only men evaluated for infertility under the age of 50, all SIRs values increased, with the highest risk is found in azoospermic men. A Cox regression model (controlling for age and year of evaluation) comparing azoospermic vs. non-azoospermic, revealed an HR = 2.2 (95% CI: 1.0–4.8)
Grasso, C. et al. [50]	Case-control study (Italy)—3b	245 cases of TC and 436 controls	Number of children fathered 5 years before diagnosis	OR per additional child = 0.78 (95% CI: 0.58–1.04)	The sibship size of the cases was inversely associated with the risk of testicular cancer **(OR = 0.76, 95% CI: 0.66–0.88**, per sibling) Age at first attempt to conceive and the association between a combined indicator of fertility (time to conception and use of assisted reproduction techniques) were not associated with the risk of testicular cancer The authors did not find evidence of heterogeneity between seminoma and nonseminomatous TC Genotyping of the KITLG single nucleotide polymorphism (SNP) rs995030 showed a strong association with the risk of testicular cancer (**OR = 1.83; 95% CI: 1.26–2.64**), but it did not modify the association between number of children and the risk of testicular cancer
Focused on several type of cancers (including data on risk of TC)					
Eisenberg, M.L. et al. [51]	Retrospective Cohort study (USA)—2b	76,083 infertile men (of them 47,385 had a male factor) vs. 112,655 who underwent vasectomy vs. 760,830 control men (matched on age and follow-up time) compared to age adjusted SEER estimates	Diagnosis and treatment infertility codes in US claims data	**SIR for TC in infertile men = 1.71 (95% CI: 1.28–2.25)** **SIR for TC in infertile men with a male factor infertility = 1.79 (95% CI: 1.24–2.50)** SIR for TC in vasectomy group= 1.21 (95% CI: 0.90–1.58) **SIR for TC in control men = 0.84 (95% CI: 0.74–0.95)**	When comparing testicular risk among three cohorts using a Cox regression models (adjusted for age, year of evaluation, comorbidity, and follow up time): **Infertile vs. control: HR = 1.99 (95% CI: 1.47–2.70)** **Infertile vs. vasectomy: HR = 1.50 (95% CI: 1.01–2.22)** Vasectomy vs. control: HR = 1.27 (**95% CI:** 0.94–1.73) Infertile men also had a higher risk of non-Hodgkin’s lymphoma and all cancers compared to both vasectomized and control men.
Hanson, H.A. et al. [52]	Retrospective Cohort study (USA)—2b	20,433 infertile men vs. 20,433 fertile control subjects matched by age and birth year	Men with semen analysis	**Cox proportional hazard regression models: HR = 3.3 (95% CI: 1.6–6.9)**	In the principal result, men with semen analysis are assumed as infertile men When compared with fertile control, men with semen analysis resulted Normozoospermic: **HR = 2.9 (95% CI: 1.2–6.7)** Oligozoospermic based on concentration (<15 M/mL) and based on count (<39 M) have **HR = 11.9 (95% CI: 4.9–28.8)** and **HR = 10.3 (95% CI: 4.1–26.2)**, respectively Hyperzoospermic: no significant difference in risk Excluding azoospermic men, increased TC risk was found as motility, viability, and TMC declined: Motile trend: **HR = 1.3 (95% CI: 1.1–1.5)** Viability Trend **HR = 1.3 (95% CI: 1.1–1.5)** TMC trend: **HR = 1.3 (95% CI: 1.1–1.6)** Head morphology, only men in the lowest quartile **(HR = 4.2)** Tail morphology, only men in the second quartile **(HR = 5.3)** Among men with semen analysis Oligozoospermic vs. normozoospermic men **HR = 3.8 (95% CI: 1.6–8.7)**. Trend tests showed a significant increase in TC risk with a decline in sperm concentration categories. A similar pattern existed with sperm count. Hyperzoospermic men vs. normozoospermic: similar risk

LE: Level of Evidence based on Oxford Centre for Evidence-Based Medicine (March 2009). TC: Testicular Cancer; CI: Confidence Interval; USA: United States of America: SEER: Surveillance, Epidemiology, and End Results; HR: Hazard Ratio; OR: Odds Ratio; RR: Relative Risk; SIR: Standardized Incidence Ratio. **Bold:** results considered statistically significant.

**Table 3 medicina-59-01305-t003:** Studies on testicular lesions found at US and clinical exams in populations of patients with infertility issues.

Reference	Population Examined, n	Population Description	Testis Focal Lesions Found % (n/Population Sampled)	Diagnosis % (n/Testis Focal Lesion)	Palpable % (Palpable/Total Lesion)	Diagnosis % (n/Palpable Identified)	Nonpalpable, % (Nonpalpable/Total Lesion)	Diagnosis % (n/Nonpalpable Identified	Nonpalpable Incidental ^#^	Nonpalpable with Other Risk Factors for TC
Retrospective studies on consecutive and nonconsecutive series of US examinations										
Patel, P.J. et al. [114]	200	Male patients who attended the infertility clinic	0% (0/200)	-	-	-	-	-	-	-
Buckspan, M.B. et al. [115]	400	Oligospermic men	1% (4/400)	BT = 4 (100%)	0% (0/4)	-	100% (4/4)	BT (4 (100%)	0	4 (100%)
Pierik, F.H. et al. [116]	1372	Infertile men	0.51% (7/1372)	M: 2 (28.6%) BT: 5 (71.4%)	14.3% (1/7)	NR	85.7% (6/7)	NR	4 (66.6%)	C: 2 (33.3%)
K Jarvi, A.Z. et al. [117] data report by [118]	450	Infertile men	NF	NF	NF	NF	3	BT/BL: 3 (100%)	NF	NF
Carmignani, L. et al. [119]	462	Infertility	1.51% (7/462)	M: 2 (28.6%) BT: 3 (42.8%) BL: 2 (28.6%)	57.1% (4/7))	M: 2 (50%) BT:1 (25%) BL: 1 (25%)	42,9% (3/7)	BT: 2 (66.6%) BL: 1 (33.3%)	3 (100%)	0 (0%)
Carmignani, L. et al. [120]	560	Azoospermic	1.42% (8/560)	M: 2 (25.0%) BT: 3 (37.5%) Bl: 3 (37.5%)	50% (4/8)	M: 2 (50%) BT: 1 (25%) BL: 1 (25%)	50% (4/8)	BT: 2 (50%) BL: 2 (50%)	3 (75%)	C = 1 (25%)
Sakamoto, H. [121]	545	Infertile men	0.18% (1/545)	NO: 1 (100%)	0% (0/1)	-	100% (1/1)	NO: 1 (100%)	1 (100%)	0 (0%)
Phillips, N. et al. [122]	749	Men attending an infertility clinic	1.2% (9/749)	M:5 (55.5%) BT:1 (11.1%) BL: 3 (33.3%)	0% (0/9)	-	100% (9/9)	M:5 (55.5%) BT:1 (11.1%) BL: 3 (33.3%)	8 (88.9%)	K = 1 (11.1%)
Eifler, J.B., Jr. et al. [118]	145	Infertile men with non-obstructive azoospermia candidate for TESE	33.7% (49/145)	M: 1 (2.0%) BT: 4 (8.16%) BL: 38 (77.5%) NO: 6 (12.2%)	42.9% (21/49)	NR	57.1% (28/49)	NR	NR	NR
Negri, L. et al. [33]	2172	Male members of infertile couples	0.41% (9/2172)	M: 4 (44.4%) BT: 3 (33.3%) BL: 2 (22.2%)	11.1% (1/9)	M:1 (100%)	88.8% (8/9)	M: 3 (37.5%) BT: 3 (37.5%) BL: 2 (25%)	7 (87.5%)	C = 1 (12.5%)
Bieniek, J.M. et al. [123]	4088	Scrotal ultrasound completed for male infertility evaluation	NR	NR	NR	NR	120	M: 6 (5%) BT–BL: 12 (10%) NO: 102 (85%)	NR	NR
Nashan, D. et al. [124]	658	Consecutive patients attending an infertility clinic	3.03% (20/658)	M: 4 (20%) BT/BL/NO: 16 (80%)	NR	NR	NR	NR	NR	NR
Behre, H.M. et al. [125] Abstract only	1048	Consecutive patients attending the Institute of Reproductive Medicine	0.47% (5/1048)	NF	NF	NF	NF	NF	NF	NF
Mancini, M. et al. [30]	145	Azoospermic patients	7.58% (11/145)	M: 2 (18.1%) BT: 3 (27.3%) BL: 4 (36.4%) NO: 2 (18.1%)	18.1% (2/11)	M: 1 (50%) BT: 1 (50%)	81.1% (9/11)	M: 1 (11.1%) BT: 2 (22.2%) BL: 4 (44.4%) NO: 2 (22.2%)	NR	NR
Prospective controlled studies										
Qublan, H.S. et al. [126]	234	234 infertile vs. 150 normospermic fertile men	0% (0/234)	-	-	-	-	-	-	-

M = Malignancy, BT = Benign Tumours; BL = Benign lesion; NO = No histological diagnosis. ^#^ Incidental = non-palpable testicular lesion without any risk factor (from medical history, clinical signs, or laboratory findings) for TC, except for male infertility. C = Cryptorchidism, KS = Klinefelter Syndrome, HTC: history of testis cancer. NR = Not reported in the original study; NF = Not found (data reported by other reference).

## Data Availability

Data reported in this review are all available at locations cited in the reference section.

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
