# Peer review of "Male Infertility and the Risk of Developing Testicular Cancer: A Critical Contemporary Literature Review"

_medicina, 2023, doi:10.3390/medicina59071305_

Round 1
Reviewer 1 Report
Giuseppe Maiolino et al. summarized the research on the association between male infertility and the risk of TC. This work provides a comprehensive review of current evidence on the effect of male infertility on TC risk, based on histopathological and epidemiological studies. The authors also discussed the risk factors of TC and provided reasonable recommendations on screening and surveillance for the infertility populations. The manuscript is well-organized and the tables are transparent and informative.
1. Lines 178-180, the author mentioned a “meta-analysis reported a 1.9-fold higher risk of testicular cancer in men with male infertility compared to men considered fertile”. The original reference of the meta-analysis should be cited.
2. On page 19, the authors pointed out the higher incidence of testicular nodules/mass detected by ultrasound in infertility. Ultrasound is currently the primary examination for TC. However, for these patients who have detectable lesions, mpMRI could be valuable in many cases, as it has much higher sensitivity and specificity in differentiating benign from malignant intratesticular lesions (AJR Am J Roentgenol. 2010 Mar;194(3):682-9. Andrology. 2021 Sep;9(5):1395-1409.). I think a brief discussion on whether and when mpMRI should be conducted for infertility populations with intratesticular lesions would be helpful.
3. Lines 19: “AA” should be “A”.
4. Many decimal points were wrongly typed as “,” such as line 90. Please revise them.
Author Response
First of all, thanks for all suggestions reported.
- We have cited the meta-analysis (lines 180 and 642-643)
- We have added a section on the use of MR on testis lesions (lines 469-492)
- We have modified the mistake in line 19.
- We have modified all decimal separators in the manuscript with a “point”.
Reviewer 2 Report
In this manuscript, Maiolino et al. reported the results of a narrative review aimed to identify and summarize the available evidence about the relationship between male infertility and the risk of developing testicular cancer.
The review is well-structured and well-organized. The Authors reported the results of multiple studies deemed relevant to the aim of the review.
Here I report my revisions and comments:
-English language editing is necessary.
-Several typo errors are present in the text. For example, line 19 (“A” and not “AA”) or line 85 (“GCNIS” and not “GNIS”). Please, double-check for the possible presence of other typo errors in all the text.
-Tables are not clear. I suggest using the “landscape” orientation to have more space to display each Table. In addition, I suggest adding the first author's name in the first column close to the study's reference. Finally, I suggest summarizing the information reported in the Tables (e.g., using more abbreviations).
-I suggest adding a Figure.
-I suggest adding a brief “Discussion” paragraph where the Authors summarize and speculate about the results of the review.
Moderate editing of English language required
Author Response
First of all, thanks for all suggestions reported.
- We have revised the English language of the manuscript (thanks to the help of one of our collaborators who works as a professional translator).
- We have checked all typo errors found in the manuscript.
- About Tables. For the landscape orientation we spoke to the publisher about it and modified the orientation. We added the first author's name and summarized the information reported in the Tables.
- We cannot add a figure for this narrative review. Our aim is to provide a broad perspective of a topic. We have no ideas about a figure to add.
- The section “Conclusion” is written with our summary and speculations about the results of the review. In narrative reviews, authors in medical literature suggested to discuss results from the studies in each corresponding paragraphs and we have followed this rule.
Round 2
Reviewer 2 Report
The Authors addressed adequately my revisions and comments.
Minor editing is necessary.